# Parent-reported child's close contact with non-household family members and their well-being during the COVID-19 pandemic: A cross-sectional survey

Lisa Woodland[1,2]*, Louise E. Smith[1,2], Samantha K. Brooks[1,2], Rebecca K. Webster[3], Richard Amlôt[2,4], Antonia Rubin[5], G. James Rubin[1,2]

1 Department of Psychological Medicine, King's College London, London, United Kingdom, 2 NIHR Health Protection Research Unit in Emergency Preparedness and Response, Bristol, United Kingdom, 3 Department of Psychology, University of Sheffield, Sheffield, United Kingdom, 4 Behavioural Science and Insights Unit, UK Health Security Agency, Salisbury, United Kingdom, 5 Trustee at Weald of Kent Grammar School, Tonbridge, Kent, United Kingdom

* lisa.woodland@kcl.ac.uk

**Data Availability Statement:** The data set and survey questions related to this study can be accessed via the King's College London research

## Abstract

In England (UK), at the start of the COVID-19 pandemic the public were required to reduce their physical contacts to slow the spread of COVID-19. We investigated the factors associated with children having: 1) close contact with family members from outside their household ('non-adherent behaviour'); and 2) low well-being (Revised Child Anxiety and Depression Scale). We conducted an online cross-sectional survey, completed at any location of the participant's choice between 8 and 11 June 2020 in parents (n = 2,010) who were aged eighteen years or over and had a school-aged child (4–18 years old). Parents reported that 15% (n = 309) of children had non-adherent contact and that 26% (n = 519) had low well-being. We used a series of binary logistic regressions to investigate associations between outcomes and child and parent characteristics. Children had higher odds of having non-household contact when they had special educational needs [adjusted odds ratio, 2.19 (95% CI, 1.47 to 3.27)], lower well-being [2.65 (95% CI, 2.03 to 3.46)], were vulnerable to COVID-19 [2.17 (95% CI, 1.45 to 3.25)], lived with someone who was over 70 years old [2.56 (95% CI, 1.55 to 4.24)] and their parent had low well-being [1.94 (95% CI, 1.45 to 2.58)]. Children had higher odds of lower well-being when they had special educational needs [4.13 (95% CI, 2.90 to 5.87)], were vulnerable to COVID-19 [3.06 (95% CI, 2.15 to 4.36)], lived with someone else who was vulnerable to COVID-19 [2.08 (95% CI, 1.64 to 2.64)], or lived with someone who was over 70 years old [2.41 (95% CI, 1.51 to 3.83)]. Many children came into contact with non-household family members, mainly for childcare. Factors relating to COVID-19, children's well-being and education were also important. If school closures are needed in future, addressing these issues may help reduce contact.

data repository, KORDS (DOI 10.18742/21757232). For further information contact: kcl - research.data research.data@kcl.ac.uk.

**Funding:** Funding: This study was funded by the Economic and Social Research Council [grant number ES/P000703/1] and by the National Institute for Health and Care Research Health Protection Research Unit (NIHR HPRU) in Emergency Preparedness and Response [grant number NIHR200890], a partnership between the UK Health Security Agency, King's College London and the University of East Anglia. The views expressed are those of the author(s) and not necessarily those of the NIHR, UKHSA or the Department of Health and Social Care. For the purpose of open access, the author has applied a Creative Commons Attribution (CC BY) licence to any Author Accepted Manuscript version arising Role of the funding source: The funders had no role in study design, data collection and analysis, decision to publish, or preparation of the manuscript. The corresponding author had full access to all the data and had final responsibility for the decision to submit for publication.

**Competing interests:** I have read the journal's policy and the authors of this manuscript have the following competing interests: G.J.R., R.A. and L.S. participate in the UK's Scientific Advisory Group for Emergencies, or its subgroups. RA is an employee of the UK Health Security Agency, which is an arm of the UK Government. This does not alter our adherence to PLOS ONE policies on sharing data and materials. These groups did not fund the study or authors.

## Introduction

On 23 March 2020, England went into 'lockdown,' with the public required to stay at home to slow the spread of COVID-19 [1]. The public were only allowed to leave home for a limited number of reasons, which included for medical attention, to buy necessities (as infrequently as possible), for exercise (only once a day outside their home) and to work if absolutely necessary and they were unable to work from home. Physical distancing guidelines were implemented: people from different households were not allowed to meet and individuals were asked to stay two meters away from people not in their household. Non-essential shops were closed, and schools were closed to most children. Children could attend school if they were on a 'health care plan' due to a specific medical or social need or if their parent(s) were employed in jobs that were essential to the COVID-19 response ('key workers'), such as doctors, teachers, and supermarket employees [2].

After two months of lockdown guidance, the restrictions started to ease. On 11 May it was announced that people could return to work, the restrictions on spending time outside were lifted, and individuals were allowed to arrange meetings with one person from another household as long as this happened outside and a two meter distance was maintained [3]. From 1 June 2020, the rules relaxed further: six people from different households were able to meet outside (though still at a two meter distance) and more children were eligible to attend school, including children in early years (four to five years), year one (five to six years) and year six (10 to 11 years) [4].

While all the restrictions caused strain within society, the closure of schools was particularly problematic. A systematic review suggested that between 18% and 60% of children scored above risk thresholds for distress, anxiety and depressive symptoms between February and July 2020 and increases in other adverse behaviours were also reported, such as an increase in suicidal ideation, worsening of mood, emotional difficulties and conduct problems in children [5]. In a qualitative study that we conducted in April 2020 we found that parent's and children's characteristics (e.g., parent's work status, children's age, access to resources for home-schooling and entertainment activities and space inside and outside the home to exercise) impacted how well families could cope with the COVID-19 school closures [6].

Determining, at the height of a crisis, whether the epidemiological benefits of school closures in terms of reducing disease transmission will exceed the psychological, physical, educational, and social costs is a daunting challenge for those who must make this decision. One question that must be factored into decision making is 'where do children go when their school is shut?'. In an ideal world (from an epidemiological perspective), parents would be able to ensure that children are placed under appropriate alternative supervision and remain apart from each other: continued mixing between children and other households would defeat the purpose of school closures. A systematic review of unplanned school closures prior to the COVID-19 pandemic found a 65% reduction in the mean total number of contacts for each student whilst schools were closed [7]. However, all 19 studies included in the review reported that children continued to meet with people from other households to some extent. A common reason for meeting others related to childcare. For example, one study found that children in households where special childcare arrangements were needed during the closure had significantly higher odds of leaving home than children who were more independent and able to take care of themselves [8]. Parental concerns about the school closure also commonly reflected the difficulties they would face in trying to arrange childcare as well as concerns about lost education [7]. Previous research about children's adherence to medical treatment and encouraging healthy eating behaviours and physical exercise follow a similar pattern; factors relating to social networks and family cohesiveness or conflict [9,10] and risk perceptions

(e.g., perceived severity and vulnerability) [11,12] are common in impacting adherence. These findings mirror a study that we conducted about family's adherence to England's COVID-19 guidelines. This study also found that parent's and children's characteristics, such as family income impacted adherence and that adherence may be more challenging for families with low psychological and physical ability [13].

In the context of the COVID-19 pandemic, issues around childcare while schools were closed were of particular concern because of a worry that grandparents might be called on to look after children in some families, putting older adults at increased risk of infection [14]. It was not just policy makers who worried about this, many children were themselves worried about their grandparents' vulnerability to COVID-19 [15] and of the risk that they themselves might infect their grandparents [16]. The extent to which these worries were borne out remains unclear.

Of course, while keeping children separate from each other and away from their grandparents might reduce disease transmission, it does present other risks. For example, a reduction in physical activity levels as a result of staying at home can affect physical and mental well-being [5]. While parents were advised to ensure their children engaged in physical activity during the initial lockdown, the extent to which this occurred, and the impact of that initial period of isolation and inactivity is uncertain. Some parents and children may also have been very cautious about the risks associated with COVID-19, engaging in additional protective behaviours that were not explicitly recommended by the Government and which may have exerted an additional toll on well-being [17,18].

In this study we investigated the factors associated with parents reporting that their children had: 1) close contact with family members from outside their household; and 2) low well-being during the first COVID-19 lockdown in the UK. Considering the evidence discussed above, we specifically explored whether close contact with non-household family members and low child well-being were associated with: parent's age, gender, and employment characteristics, children's age, gender and special educational needs status, and household's vulnerability to, perceptions about and behaviours in relation to COVID-19.

## Materials and methods

### Design

We commissioned a market research company, BMG Research [19] to administer an online cross-sectional survey between 8 and 11 June 2020. Data collection occurred after lockdown restrictions had begun to be eased. At this stage in the pandemic, schools had re-opened to children in early years (four to five years), year one (five to six years) and year six (1 to 11 years). However, physical distancing restrictions remained in place throughout society, and while up to six non-household members could meet, this had to be outside and at a two-meter distance.

We have previously published data from this survey relating to parental perceptions of the hygiene procedures within schools [20] and investigating why some parents did not send their eligible children back to school [21].

### Participants

Participants (n = 2,447) were recruited from BMG Research's panel. To achieve a sample that was broadly representative of the population, BMG Research monitored region, child age, child gender, parent/guardian age, and parent/guardian gender. Participants were eligible for the study if they were aged eighteen years or over, lived in England, and were a parent or guardian to a school-aged child (4–18 years old) who usually lived with them. One-hundred

and eighty-three participants were screened out for non-eligibility by BMG, 226 participants dropped out after starting the survey and 28 completed but were removed for reasons related to quality control, such as completing the survey quickly compared to the average (14 minutes) and median (11 minutes) survey time or for 'straight-lining' (selecting the same option for every question) suggesting inattention to the questions. These issues are common in online surveys [22]. Two thousand and ten participants remained.

Quota sampling, as used in this study, is a non-probability based approach and we have therefore not reported the response rates as they are not helpful indicators of non-response bias. The response rates for each quota will differ, the denominator for each is essentially unknown, and the predominant source of bias is in the make-up of the underlying panels that are being recruited from, rather than whether the sample is representative of the panel [23].

The sample fell within five percentage points of the national population by the child's gender, key stage, and type of school attended against the known distribution for school children in England [24].

Participants were paid in points that could be accumulated by the participant and exchanged later for money. Our survey paid points equivalent to about £0.60.

The research was approved by the Psychiatry, Nursing and Midwifery Research Ethics Sub-committee at King's College London (LRS-19/20-18787). An information sheet was provided at the start of the survey, which described the purpose of the study. Participants were also informed of the study process, and they provided written consent before taking part.

## Study materials

The full survey is available in the (S1 Text).

All participants answered questions referring to their child who had the most recent birthday. In rare cases where children shared a birthday, we asked the parent to select one child.

## Outcome one: Child's physical close contact with non-household family members

We asked parents to choose from a list of seven options about people who the child had come into close contact with in the past 24 hours ("someone [child] lives with;" "friends or other children who [child] does not live with"; "a babysitter, nanny or childminder;" "family member aged under 70 who [child] does not live with;" "family member aged over 70 who [child] does not live with;" "other children, not already reported above"; and "other adults, not already reported above"). We made clear in the question that "by close contact we mean closer than 2 meters, for fifteen minutes or more," which was the UK Government's definition of close contact [25]. Parents were asked to report all the options that applied.

We created a binary variable to indicate whether the child had close contact with a family member from outside their household, defined as a "non-household close contact." This included children reported as having had close contact with either "a family member aged under 70 who [child] does not live with" or "a family member aged over 70 who [child] does not live with."

## Outcome two: Child's well-being

We asked parents to report the child's well-being using two subscales from the Revised Child Anxiety and Depression Scale (RCADS): the generalised anxiety disorder (GAD) sub-scale and the major depressive disorder (MDD) sub-scale [26]. The GAD sub-scale asks parents to respond to six statements about their child (e.g., "my child worries about things"; and "my child worries that something awful will happen to someone in the family"). The MDD sub-

scale asks parents to respond to ten statements about their child (e.g., "my child feels sad or empty"; "nothing is much fun for my child"; and "my child has trouble sleeping"). Parents can respond "never," "sometimes," "often," and "always."

We created a binary variable to indicate well-being in the child. The variable was recoded using SPSS syntax supplied by the RCADS authors, which assigns a value against each answer from 0 ("never") to 3 ("always") on the GAD and MDD RCADS sub-scales and creates a total score for each sub-scale [26]. The total score is turned into a t-score, normalising the RCADS scores within the population, by child's age and gender. A t-score of 65 (approximately in the top 7% of un-referred young people of the same age) on either the GAD or MDD sub-scale was our well-being cut off. "Lower well-being" represents a child with a medium or severe risk of clinical mental illness and "higher well-being" represents a child with low risk of clinical mental illness.

## Predictor variables: Parent and child personal characteristics

We asked parents to report their gender, age, region, household income, employment status, and if employed, whether they were working from home, level of education, marital status, ethnicity, and key worker status. We asked parents to report the child's gender, age, school year, and whether the child had special educational needs. We also asked whether anyone within the household was aged over 70 years old or had a health condition that made them vulnerable to COVID-19 and whether they had access to outside space.'

We recoded household income (less than £5,000; £5,000-£9,999; £10,000-£14,999; £15,000-£19,999; £20,000-£24,999; £25,000-£29,999; £30,000-£34,999; £35,000-£39,999; £40,000-£44,999; £45,000-£49,999; £50,000-£59,999; £60,000-£69,999; £70,000-£84,999; £85,000-£99,999; more than £100,000), employment status (full time paid job (31+ hours); part time paid job (<31 hours); doing paid work on a self-employed basis or within your own business; employed, but currently furloughed; student / on a government training programme (Nation Traineeship/ Modern Apprenticeship); out of work (6 months or less); out of work (more than 6 months); looking after home / homemaker; retired; disabled OR long-term sick; unpaid work for a business, community or voluntary organisation), parent education level (PhD/Doctor; Master's; Bachelor's Degree or equivalent (such as a NVQ level 5); higher education (such as a HND or a NVQ level 4); A-level or equivalent (such as Scottish Highers or NVQ level 3); GCSE and below (such as O level or an RSA Diploma); Other qualifications (Such as NVQ level 1); No qualifications), marital status (single (i.e. never married and never registered as a same sex civil-partnership); co-habiting with partner (but never married or been in a civil partnership); civil partnership; married; separated, but still legally married / in a civil partnership; divorced / civil partnership legally dissolved; widowed / surviving partner from a same-sex civil partnership), and ethnicity (English/Welsh/Scottish/Northern Irish/British; Irish; Gypsy or Irish Traveller; White and Black Caribbean; White and Black African; White and Asian; Indian; Pakistani; Bangladeshi; Chinese; Caribbean; African; Arab; any other (please specify)) into binary variables.

We recoded parent age, key worker status, and access to outside space into categorical variables, as shown in the results tables. We recoded child school year into Key Stages as used in the English education system (Early years = ages 4 to 5; Key Stage 1 = ages 5 to 7; Key Stage 2 = ages 7 to 11; Key Stage 3 = ages 11 to 14; Key Stage 4 = ages 14 to 16; Years 12 and 13 = ages 16 to 18). We created two binary variables to indicate whether the child, and someone in the household (other than the child) had a health condition that might make them particularly vulnerable to COVID-19. The responses "yourself [participant]" and "anyone else you live with" were combined into one variable to indicate that the child lived with someone vulnerable to COVID-19.

## Parent's well-being

We asked parents to report their well-being using the Patient-Health Questionnaire-4 (PHQ4) [27], which asks "over the last two weeks, how often have you been bothered by the following problems:" "feeling nervous, anxious or on the edge;" "not being able to stop or control worrying;" "little interest or pleasure in doing things;" and "feeling down, depressed, or hopeless." Parents can respond "not at all," "several days," "more than half the days," and "nearly every day."

We created a binary variable to indicate low well-being in the parent. We assigned a value against each answer from 0 ("not at all") to 3 ("nearly every day") on the PHQ4 and summed responses (range 0 to 12). We used a cut off score of 5 or above to indicate low well-being in the parent, indicating moderate or severe risk of clinical anxiety or depression.

## Child's activities outside the home

We asked parents how many times the child had left the home in the past seven days: "to go to the shops for groceries, toiletries, or medicines;" "to go to the shops for other items;" "for exercise;" "for a medical need (e.g., an outpatient appointment);" "to go to school;" "to provide help to someone else;" "to meet friends;" to meet family members who they did not live with; and "for another reason."

Items about the child's activities outside the home were used as continuous variables.

## Behaviours that parents and children had followed

We asked parents to report the behaviours that they or their children had followed in the past 7 days because of the risk of COVID-19 (e.g., "washed your hands thoroughly and regularly," "stayed 2m (3 steps) away from people you do not live with when outside your home," "washed your clothes when you have returned home and "washed [child]'s clothes when she/he has returned home"). Parents could respond "yes" or "no" to each statement. Out of the 11 statements that we asked about, two were recommended and nine were not recommended by UK Government at the time of the study.

We created a continuous variable to indicate the COVID-19 behaviours that parents and their child had followed by combining all 11 responses to the statements about what the parent or child had done in the past seven days because of the risk of COVID-19.

## Statements about lockdown

We included 16 statements about lockdown, which included questions about COVID-19 (e.g., "if [child] goes out, she/he is likely to catch coronavirus"), schooling (e.g., "[child] is keeping up with his/her schoolwork") and home environment (e.g., "in the past 7 days, [child] has been bored"). Parents responded to each statement using a five-point Likert-scale from "strongly agree" to "strongly disagree," or "not applicable."

Items about lockdown were used as continues variables.

## Missing data

We took the pragmatic approach to code the responses "don't know", "not applicable", "prefer not to say" and "prefer to self-describe" as missing data. We adopted this approach for outcome variables because it was not possible to categorise participants as adherent or not, or as having good well-being or not. For predictors we adopted this approach to provide a clearer understanding of the difference between endorsing, or not endorsing, each variable in terms of its impact on our outcomes.

### Validity and reliability

Previous studies have used RCADS [28], and the PHQ-4 [29], and they have both demonstrated good validity and reliability. The survey questions that related to COVID-19 were designed for this study based on our previous research [6], they have face validity and we copied the wording of official Government guidance where applicable. Five parents and one school trustee (see public involvement section) piloted the survey. Beyond this we do not have psychometric data about the validity and reliability of the questions.

### Public involvement

A school trustee contributed to the development of the survey materials and co-authored this paper. The survey questions were reviewed by five parents of school children who also piloted the survey before publication. The feedback we received resulted in minor changes to the wording of some survey questions for clarity.

### Analysis

We ran a series of binary logistic regressions using SPSS [30,31] investigating the univariable associations between our two outcomes and each of our predictor variables. We ran a second set of binary logistic regressions controlling for child and parent characteristics (participant gender, age, region, household income, employment status, education level, marital status, ethnicity, and the child's gender and school year). These 'pre-exposure' variables were selected based on previous research [6,13,21] that indicated these variables may be associated with exposure and / or outcome variables, and these variables could be measured, which is recommended when causation is unknown [32].

We applied a Bonferroni correction to our results (p≤0.001) due to running many analyses. Only associations that met this level are discussed narratively in the results section.

## Results

Most parents were: female (53%); between 36 and 45 years of age (43%); lived in London (17%); had a household income over £35,000 (55%); working (83%); highly educated (57%); married or co-habiting (84%); and of white ethnicity (87%) (Table 1).

### Factors associated with children's non-household close contact

Of the 2010 parent responses, 15% (95% confidence interval (CI), 14% to 17%, n = 309) of children were perceived by the parents to have had close contact with a family member that they did not live with in the past 24 hours. This included 9% (n = 189) who had close contact with a family member aged under 70 years and 6% (n = 120) who had close contact with a family member aged 70 years or older, which includes n = 45 who had close contact with non-household family members under and over 70 years of age. The parent and child characteristics associated with children who had non-household close contact are shown in Table 2 (see Table 7, Appendix A for the descriptives in S1 Appendix). Parents with lower well-being had higher odds of reporting that their child had non-household close contact [adjusted odds ratio 1.94 (95% CI, 1.45 to 2.58)]. Parents who reported that their child had special educational needs, lower well-being, were vulnerable to COVID-19, and lived with someone who was over 70 years of age had higher odds of reporting that their child had had non-household close contact [adjusted odd ratios, 2.19 (95% CI, 1.47 to 3.27); 2.65 (95% CI, 2.03 to 3.46); 2.56 (95% CI, 1.55 to 4.24), respectively].

Table 3 shows the associations between children who had non-household close contact, as perceived by their parents, and the predictor variables relating to parent perceptions about

**Table 1. Parent, child and household characteristics (n = 2,010).**

| Variable | Level | n, (%)* |
|---|---|---|
| Parent gender | Male | 931 (46) |
| | Female | 1065 (53) |
| Parent age | 18–35 years | 394 (20) |
| | 36–45 years | 868 (43) |
| | 46 years ≥ | 748 (37) |
| Region | East Midlands | 151 (8) |
| | East of England | 219 (11) |
| | North East | 118 (6) |
| | North West | 273 (14) |
| | South East | 335 (17) |
| | South West | 163 (8) |
| | West Midlands | 211 (10) |
| | Yorkshire & the Humber | 195 (10) |
| | London | 345 (17) |
| Household income | ≤ £34,999 | 804 (40) |
| | £35,000 ≥ | 1114 (55) |
| Employment status[1] | Working | 1677 (83) |
| | Not working | 321 (16) |
| Parent Working from home[2] | Yes | 940 (59) |
| | No | 628 (40) |
| Education level | ≤ A-level | 855 (43) |
| | Degree ≥ | 1139 (57) |
| Marital status | Living alone | 327 (16) |
| | Married/cohabiting | 1683 (84) |
| Ethnicity | White | 1753 (87) |
| | BAME | 241 (12) |
| Key worker status | Both parents | 195 (10) |
| | One parent | 847 (42) |
| | No | 956 (48) |
| Child gender | Boy | 1063 (53) |
| | Girl | 947 (47) |
| Child school year | Early Years (ages 4 to 5) | 101 (5) |
| | Key Stage 1 (ages 5 to 7) | 356 (18) |
| | Key Stage 2 (ages 7 to 11) | 675 (34) |
| | Key Stage 3 (ages 11 to 14) | 395 (20) |
| | Key Stage 4 (ages 14 to 16) | 328 (16) |
| | Years 12 & 13 (ages 16 to 18) | 155 (8) |
| Child had special educational needs | Yes | 161 (8) |
| | No | 1832 (91) |
| Child lower well-being | Yes | 519 (26) |
| | No | 1491 (74) |
| Parent lower well-being | Yes | 391 (19) |
| | No | 1619 (81) |
| Child vulnerable COVID-19 | Yes | 157 (8) |
| | No | 1826 (91) |
| Household vulnerable COVID-19 | Yes | 510 (25) |
| | No | 1311 (65) |

(*Continued*)

**Table 1.** (Continued)

| Variable | Level | n, (%)* |
|---|---|---|
| Someone over 70 years | Yes | 87 (4) |
| | No | 1923 (96) |
| COVID-19 behaviours followed by child and parent | Continuous (0 = followed no behaviours, 11 = followed all behaviours) | N = 2,010, M = 5.60, SD = 3.10 |
| Access to outside space[3] | Garden | 1784 (89) |
| | Other[3] | 157 (8) |
| | No | 69 (3) |

*Percentages may not total 100 due to rounding errors and missing data.

[1] Working includes students and volunteers.

[2] Question only offered to participants who reported working in a paid job (full-time, part-time, and self-employed) and not to participants who reported being a student, on furlough and a volunteer.

[3] Participants that reported no access to a garden but had access to other outdoor spaces such as patio, terrace, and balcony.

Abbreviations: N = number; % = percentage; M = mean; SD = standard deviation.

lockdown (see Table 8, Appendix B for the descriptives in S1 Appendix). Parent's agreement that their child had extra support at school before the closures, were upset about not seeing other family members that they did not live with, and that the parent had found it hard to keep up with work or other important commitments was associated with their children's non-household close contact [adjusted odd ratios, 0.82 (95% CI, 0.74 to 0.90); 0.84 (95% CI, 0.75 to 0.93); 0.78 (95% CI, 0.70 to 0.87), respectively].

The behaviours that families followed, perceived by parents because of the risk of COVID-19 are presented in (see Table 9, Appendix C in S1 Appendix).

## Factors associated with children's lower well-being

Of the 2010 parent responses, 26% (95% CI, 24% to 28%, n = 519) reported that their child was perceived to have had low well-being. The parent and child characteristics associated with child low well-being are shown in Table 4 (see Table 10, Appendix D for the descriptives in S1 Appendix). Parents aged between 18 and 35 years old, who were a key worker and parents with lower well-being had higher odds of reporting that their child had lower well-being [adjusted odds ratio, 1.92 (95% CI, 1.40 to 2.64); 1.51 (95% CI, 1.20 to 1.90); 7.26 (95% CI, 5.62 to 9.38), respectively]. Parents who reported that their child had had special educational needs, were vulnerable to COVID-19, lived with someone else who was also vulnerable to COVID-19, and lived with someone that was over 70 years old also had higher odds of reporting that their child had lower well-being [adjusted odd ratios, 4.13 (95% CI, 2.90 to 5.87)]; 3.06 (95% CI, 2.15 to 4.36); 2.08 (95% CI, 1.64 to 2.64)]; 2.41 (95% CI, 1.51 to 3.83), respectively]. Parents who reported that their household followed multiple precautionary behaviours because of the risk of COVID-19 had higher odds of reporting that their child had lower well-being [adjusted odds ratio, 1.11 (95% CI, 1.07 to 1.15)]. All lockdown statements bar two were associated with lower well-being (Table 5), (see Table 11, Appendix E for descriptives in S1 Appendix).

## Associations between well-being and leaving home

Parents who perceived that their child had left the home for shopping, to provide help to someone else, to meet family, for medical treatment or for another reason had higher odds of

**Table 2. Binary logistic regression comparing parent and child characteristics and associations with children's non-household close contact.**

| Variable | Level | Unadjusted odds Ratio (95% CI) | P-value | Adjusted odds ratio (95% CI) † | P-value |
|---|---|---|---|---|---|
| Parent gender | Male | 0.88 (0.69 to 1.12) | 0.30 | 0.97 (0.74 to 1.27) | 0.80 |
|  | Female | Reference |  | Reference |  |
| Parent age | 18–35 years | **2.13** (1.54 to 2.95)** | **<0.001** | **1.28* (1.25 to 2.65)** | **0.002** |
|  | 36–45 years | 1.29 (0.97 to 1.72) | 0.09 | 1.17 (0.85 to 1.62) | 0.34 |
|  | 46 years ≥ | Reference |  | Reference |  |
| Region | East Midlands | 0.73 (0.43 to 1.24) | 0.25 | 0.69 (0.40 to 1.21) | 0.19 |
|  | East of England | 0.76 (0.48 to 1.21) | 0.25 | 0.77 (0.47 to 1.26) | 0.30 |
|  | North East | 0.58 (0.31 to 1.08) | 0.08 | 0.55 (0.28 to 1.06) | 0.07 |
|  | North West | 0.76 (0.50 to 1.17) | 0.21 | 0.83 (0.52 to 1.32) | 0.43 |
|  | South East | 0.79 (0.53 to 1.18) | 0.25 | 0.84 (0.55 to 1.28) | 0.43 |
|  | South West | 0.60 (0.35 to 1.03) | 0.07 | 0.61 (0.35 to 1.08) | 0.09 |
|  | West Midlands | 0.66 (0.41 to 1.07) | 0.09 | 0.63 (0.37 to 1.05) | 0.08 |
|  | Yorkshire & the Humber | 0.91 (0.58 to 1.44) | 0.69 | 0.88 (0.54 to 1.44) | 0.61 |
|  | London | Reference |  | Reference |  |
| Household income | ≤ £34,999 | **1.34* (1.05 to 1.72)** | **0.02** | **1.44* (1.08 to 1.92)** | **0.01** |
|  | £35,000 ≥ | Reference |  | Reference |  |
| Employment status[1] | Working | 0.27 (1.21 to 0.86 to 1.72) | 0.27 | 1.12 (0.77 to 1.63) | 0.57 |
|  | Not working | Reference |  | Reference |  |
| Parent Working from home[2] | Yes | **1.53* (1.15 to 2.05)** | **0.004** | ^^^ | ^^^ |
|  | No | Reference |  | Reference |  |
| Education level | ≤ A-level | **0.77* (0.60 to 0.98)** | **0.04** | **0.70* (0.53 to 0.93)** | **0.01** |
|  | Degree ≥ | Reference |  | Reference |  |
| Marital status | Living alone | **1.46* (1.08 to 1.98)** | **0.01** | 1.36 (0.98 to 1.90) | 0.07 |
|  | Married/cohabiting | Reference |  | Reference |  |
| Ethnicity | White | 0.97 (0.67 to 1.41) | 0.88 | 1.25 (0.83 to 1.87) | 0.28 |
|  | BAME | Reference |  | Reference |  |
| Key worker status | Both parents | 1.18 (0.76 to 1.84) | 0.47 | 1.18 (0.74 to 1.90) | 0.49 |
|  | One parent | **1.65** (1.28 to 2.14)** | **<0.001** | **1.53* (1.16 to 2.02)** | **0.002** |
|  | No | Reference |  | Reference |  |
| Child gender | Boy | 1.14 (0.89 to 1.46) | 0.29 | 1.14 (0.88 to 1.48) | 1.14 |
|  | Girl | Reference |  | Reference |  |
| Child school year | Early Years | 1.53 (0.73 to 3.18) | 0.26 | 1.22 (0.55 to 2.71) | 0.63 |
|  | Key Stage 1 | **2.24* (1.28 to 3.94)** | **0.01** | **2.11* (1.13 to 3.94)** | **0.02** |
|  | Key Stage 2 | 1.40 (0.81 to 2.41) | 0.23 | 1.28 (0.70 to 2.33) | 0.42 |
|  | Key Stage 3 | 1.48 (0.84 to 2.63) | 0.18 | 1.53 (0.83 to 2.83) | 0.17 |
|  | Key Stage 4 | 1.10 (0.60 to 2.01) | 0.77 | 1.21 (0.64 to 2.30) | 0.56 |
|  | Years 12 & 13 | Reference |  | Reference |  |
| Child had special educational needs | Yes | **2.16** (1.49 to 3.14)** | **<0.001** | **2.19** (1.47 to 3.27)** | **<0.001** |
|  | No | Reference |  | Reference |  |
| Child lower well-being | Yes | **2.66** (2.07 to 3.42)** | **<0.001** | **2.65** (2.03 to 3.46)** | **<0.001** |
|  | No | Reference |  | Reference |  |
| Parent lower well-being | Yes | **2.15** (1.64 to 2.82)** | **<0.001** | **1.94** (1.45 to 2.58)** | **<0.001** |
|  | No | Reference |  | Reference |  |
| Child vulnerable COVID-19 | Yes | **2.50** (1.73 to 3.61)** | **<0.001** | **2.17** (1.45 to 3.25)** | **<0.001** |
|  | No | Reference |  | Reference |  |

*(Continued)*

**Table 2.** (Continued)

| Variable | Level | Unadjusted odds Ratio (95% CI) | P-value | Adjusted odds ratio (95% CI) † | P-value |
|---|---|---|---|---|---|
| Household vulnerable COVID-19 | Yes | 1.16 (0.87 to 1.54) | 0.32 | 1.17 (0.87 to 1.59) | 0.30 |
| | No | Reference | | Reference | |
| Someone over 70 years | Yes | **2.77** (1.74 to 4.42)** | **<0.001** | **2.56** (1.55 to 4.24)** | **<0.001** |
| | No | Reference | | Reference | |
| COVID-19 behaviours followed by child and parent | Continuous (0 = followed no behaviours, 11 = followed all behaviours) | **1.06* (1.02 to 1.11)** | **0.002** | 1.04 (0.99 to 1.08) | 0.10 |
| Access to outside space[3] | Garden | 0.73 (0.40 to 1.38) | 0.33 | 0.88 (0.44 to 1.75) | 0.71 |
| | Other[3] | 1.28 (0.63 to 2.60) | 0.49 | 1.25 (0.58 to 2.73) | 0.71 |
| | No | Reference | | Reference | |

* $p \leq \cdot 05$ and formatted bold.

** $p \leq \cdot 001$ and formatted bold.

Some results are rounded to 3 rather than 2 decimal places to distinguish between $p \leq \cdot 05$ and $p \leq \cdot 001$.

† When interpretating the predictor listed in the row, we controlled for the other variables listed here: participant gender, age, region, household income, employment status, education level, marital status, ethnicity, and the child's gender and school year.

^^^ Not included variable highly correlated with employment variable.

[1] Working includes students and volunteers.

[2] Question only offered to participants who reported working in a paid job (full-time, part-time, and self-employed) and not to participants who reported being a student, on furlough and a volunteer.

[3] Participants that reported no access to a garden but had access to other outdoor spaces such as patio, terrace, and balcony.

Abbreviations: CI = confidence interval.

reporting that their child had lower well-being (Table 6) [adjusted odds ratio, 1.24 (95% CI, 1.13 to 1.35); 1.37 (95% CI, 1.21 to 1.56); 1.19 (95% CI, 1.07 to 1.33); 3.70 (95% CI, 2.64 to 5.18); 1.22 (95% CI, 1.08 to 1.37), respectively]. Parents who perceived that their child was not leaving the home to exercise also had higher odds of reporting that their child had lower well-being [adjusted odds ratio, 0.93 (95% CI, 0.89 to 0.97)].

## Discussion

During the COVID-19 pandemic, interventions were implemented to reduce physical contacts, including school closures and limits to physical contact between members of different households. We explored the impact of these measures to identify the factors associated with children having: 1) close contact with family members from outside their household and 2) a low well-being.

### Physical close contact with non-household family members

Our finding that 15% of children had close contact with non-household family members when schools were closed is concerning, as this would have increased the risk of disease transmission. This is particularly problematic for the 6% who had close contact with a non-household family member aged 70 years or over, who would be particularly at risk from COVID-19.

We found several variables associated with non-household contact that seem to point towards increased odds for such interactions among families that require childcare. This included, for example, when one parent was a key worker and when parents reported they had been unable to keep up with work or other important commitments. At the time of the survey parents were able to go back to work and some children were eligible to attend school.

**Table 3. Binary logistic regression comparing statements about lockdown and associations with children's non-household close contact.** Close contact is defined by a child's close contact with a family member outside the household (n = 309).

| Statement | Level | Unadjusted odds ratio (95%, CI) | P-value | Adjusted odds ratio (95%, CI) † | P-value |
|---|---|---|---|---|---|
| If child goes out, she/he is likely to catch coronavirus | 5-point Likert-scale (1 = strongly agree, 5 = strongly disagree) | **0.85* (0.75 to 0.95)** | **0.004** | 0.89 (0.79 to 1.01) | 0.07 |
| If child goes out, she/he is likely to bring coronavirus back into our home | 5-point Likert-scale (1 = strongly agree, 5 = strongly disagree) | **0.86* (0.77 to 0.96)** | **0.01** | 0.91 (0.81 to 1.03) | 0.13 |
| Child is keeping up with his/her schoolwork | 5-point Likert-scale (1 = strongly agree, 5 = strongly disagree) | **1.16* (1.04 to 1.29)** | **0.01** | **1.14* (1.02 to 1.27)** | **0.03** |
| I feel confident helping child with her/his schoolwork | 5-point Likert-scale (1 = strongly agree, 5 = strongly disagree) | 1.06 (0.94 to 1.19) | 0.32 | 1.08 (0.96 to 1.23) | 0.20 |
| I feel supported by child's school | 5-point Likert-scale (1 = strongly agree, 5 = strongly disagree) | 1.00 (0.89 to 1.12) | 0.96 | 0.99 (0.68 to 1.12) | 0.87 |
| I have access to all the resources that child needs to do her/his schoolwork | 5-point Likert-scale (1 = strongly agree, 5 = strongly disagree) | 1.05 (0.94 to 1.18) | 0.38 | 1.09 (0.96 to 1.23) | 0.18 |
| During lockdown, child has learned about important things she/he wouldn't normally learn at school | 5-point Likert-scale (1 = strongly agree, 5 = strongly disagree) | 0.98 (0.87 to 1.11) | 0.74 | 1.03 (0.30 to 1.17) | 0.69 |
| In the past 7 days, child has been bored | 5-point Likert-scale (1 = strongly agree, 5 = strongly disagree) | **0.84** (0.75 to 0.93)** | **0.001** | **0.85* (0.76 to 0.95)** | **0.004** |
| In the past 7 days, my household has had a regular structure to the day | 5-point Likert-scale (1 = strongly agree, 5 = strongly disagree) | 1.12 (1.00 to 1.26) | 0.05 | 1.09 (0.96 to 1.23) | 0.18 |
| In the past 7 days, child has kept in touch with her/his friends | 5-point Likert-scale (1 = strongly agree, 5 = strongly disagree) | 1.00 (0.90 to 1.12) | 0.96 | 0.95 (0.84 to 1.07) | 0.36 |
| Child is worried about coronavirus | 5-point Likert-scale (1 = strongly agree, 5 = strongly disagree) | 0.98 (0.88 to 1.09) | 0.72 | 0.99 (0.88 to 1.10) | 0.82 |
| In the past 7 days, child has felt upset about not seeing other family members who do not live with us | 5-point Likert-scale (1 = strongly agree, 5 = strongly disagree) | **0.81** (0.73 to 0.89)** | **<0.001** | **0.84** (0.75 to 0.93)** | **0.001** |
| In the past 7 days, I have found it hard to keep up with work or other important commitments | 5-point Likert-scale (1 = strongly agree, 5 = strongly disagree) | **0.75** (0.67 to 0.83)** | **<0.001** | **0.78** (0.70 to 0.87)** | **<0.001** |
| In the past 7 days, people in my household have been getting along well | 5-point Likert-scale (1 = strongly agree, 5 = strongly disagree) | 1.13 (1.00 to 1.28) | 0.05 | 1.07 (0.94 to 1.22) | 0.28 |
| I am worried about the financial impact of lockdown measures | 5-point Likert-scale (1 = strongly agree, 5 = strongly disagree) | 0.92 (0.83 to 1.02) | 0.13 | 0.93 (0.83 to 1.03) | 0.18 |
| Before the school closures, child had extra support at school | 5-point Likert-scale (1 = strongly agree, 5 = strongly disagree) | **0.80** (0.73 to 0.88)** | **<0.001** | **0.82** (0.74 to 0.90)** | **<0.001** |

\* $p \leq \cdot 05$ and formatted bold.

\*\* $p \leq \cdot 001$ and formatted bold.

Some results are rounded to 3 rather than 2 decimal places to distinguish between $p \leq \cdot 05$ and $p \leq \cdot 001$.

† When interpretating the predictor listed in the row, we controlled for the other variables listed here: participant gender, age, region, household income, employment status, education level, marital status, ethnicity, and the child's gender and school year.

Abbreviations: CI = confidence interval.

However, research suggests that most eligible children were still not attending school [21], and many will have needed alternative childcare.

We found that the odds for non-household family close contact was relatively equal across all children's ages, which contrasts with previous research [7]. We suspect that the restrictions in place at the time of our survey explains this result. A longitudinal German study found that during COVID-19, the activities that children engaged in differed between ages [33]. However, these age differences disappeared when the guidance was more restrictive. That study also found that children commonly met elderly relatives (and friends) throughout the one-year study period, which supports our findings.

**Table 4. Binary logistic regression comparing parent and child characteristics and associations with children's lower well-being (n = 519).**

| Variable | Level | Unadjusted odds Ratio (95% CI) | P-value | Adjusted odds ratio (95% CI) † | P-value |
|---|---|---|---|---|---|
| Parent gender | Male | 0.91 (0.74 to 1.11) | 0.36 | 0.92 (0.74 to 1.15) | 0.46 |
| | Female | Reference | | Reference | |
| Parent age | 18–35 years | **1.67** (1.27 to 2.19)** | **<0.001** | **1.92** (1.40 to 2.64)** | **<0.001** |
| | 36–45 years | 1.11 (0.88 to 1.40) | 0.38 | 1.25 (0.96 to 1.62) | 0.09 |
| | 46 years ≥ | Reference | | Reference | |
| Region | East Midlands | 1.27 (0.84 to 1.91) | 0.26 | 1.21 (0.78 to 1.88) | 0.39 |
| | East of England | 0.88 (0.60 to 1.29) | 0.51 | 0.93 (0.61 to 1.39) | 0.71 |
| | North East | 0.72 (0.44 to 1.18) | 0.20 | 0.77 (0.46 to 1.30) | 0.33 |
| | North West | 0.80 (0.56 to 1.15) | 0.23 | 0.90 (0.61 to 1.33) | 0.61 |
| | South East | 0.90 (0.64 to 1.26) | 0.53 | 0.90 (0.62 to 1.29) | 0.56 |
| | South West | 0.98 (0.64 to 1.48) | 0.91 | 1.04 (0.67 to 1.63) | 0.85 |
| | West Midlands | 0.79 (0.54 to 1.18) | 0.25 | 0.75 (0.49 to 1.15) | 0.19 |
| | Yorkshire & the Humber | 0.72 (0.48 to 1.09) | 0.12 | 0.71 (0.46 to 1.11) | 0.13 |
| | London | Reference | | Reference | |
| Household income | ≤ £34,999 | 1.12 (0.91 to 1.37) | 0.29 | 1.08 (0.85 to 1.37) | 0.54 |
| | £35,000 ≥ | Reference | | Reference | |
| Employment status[1] | Working | 1.12 (0.85 to 1.48) | 0.44 | 1.12 (0.82 to 1.52) | 0.48 |
| | Not working | Reference | | Reference | |
| Parent Working from home[2] | Yes | 1.16 (0.92 to 1.47) | 0.20 | ^^^ | ^^^ |
| | No | Reference | | Reference | |
| Education level | ≤ A-level | 0.92 (0.75 to 1.12) | 0.39 | 0.92 (0.74 to 1.16) | 0.50 |
| | Degree ≥ | Reference | | Reference | |
| Marital status | Living alone | 1.17 (0.90 to 1.53) | 0.24 | 1.15 (0.87 to 1.53) | 0.33 |
| | Married/cohabiting | Reference | | Reference | |
| Ethnicity | White | 0.93 (0.69 to 1.26) | 0.65 | 1.06 (0.76 to 1.48) | 0.73 |
| | BAME | Reference | | Reference | |
| Key worker status | Both parents | **1.70* (1.21 to 2.39)** | **0.002** | **1.71* (1.18 to 2.46)** | **0.004** |
| | One parent | **1.60** (1.29 to 1.98)** | **<0.001** | **1.51** (1.20 to 1.90)** | **<0.001** |
| | No | Reference | | Reference | |
| Child gender | Boy | 1.20 (0.98 to 1.47) | 0.07 | 1.22 (0.98 to 1.50) | 0.07 |
| | Girl | Reference | | Reference | |
| Child school year | Early Years | 0.99 (0.55 to 1.79) | 0.98 | 0.71 (0.38 to 1.32) | 0.28 |
| | Key Stage 1 | 1.13 (0.73 to 1.75) | 0.59 | 0.80 (0.50 to 1.29) | 0.37 |
| | Key Stage 2 | 0.97 (0.65 to 1.47) | 0.90 | 0.71 (0.46 to 1.10) | 0.13 |
| | Key Stage 3 | 1.22 (0.79 to 1.87) | 0.37 | 1.03 (0.66 to 1.61) | 1.61 |
| | Key Stage 4 | 1.36 (0.88 to 2.11) | 0.17 | 1.15 (0.73 to 1.80) | 0.55 |
| | Years 12 & 13 | Reference | | Reference | |
| Child has special educational needs | Yes | **4.24** (3.05 to 5.89)** | **<0.001** | **4.13** (2.90 to 5.87)** | **<0.001** |
| | No | Reference | | Reference | |
| Parent low well-being | Yes | **6.77** (5.33 to 8.60)** | **<0.001** | **7.26** (5.62 to 9.38)** | **<0.001** |
| | No | Reference | | Reference | |
| Child vulnerable COVID-19 | Yes | **3.34** (2.40 to 4.65)** | **<0.001** | **3.06** (2.15 to 4.36)** | **<0.001** |
| | No | Reference | | Reference | |
| Household vulnerable COVID-19 | Yes | **2.01** (1.60 to 2.51)** | **<0.001** | **2.08** (1.64 to 2.64)** | **<0.001** |
| | No | Reference | | Reference | |

(*Continued*)

**Table 4.** (Continued)

| Variable | Level | Unadjusted odds Ratio (95% CI) | P-value | Adjusted odds ratio (95% CI) † | P-value |
|---|---|---|---|---|---|
| Someone over 70 years | Yes | **2.69** (1.75 to 4.16)** | **<0.001** | **2.41** (1.51 to 3.83)** | **<0.001** |
| | No | Reference | | Reference | |
| COVID-19 behaviours followed by child and parent | Continuous (0 = followed no behaviours, 11 = followed all behaviours) | **1.11** (1.07 to 1.14)** | **<0.001** | **1.11** (1.07 to 1.15)** | **<0.001** |
| Access to outside space[3] | Garden | 0.89 (0.52 to 1.53) | 0.67 | 1.10 (0.60 to 2.03) | 0.76 |
| | Other[3] | 1.19 (0.64 to 2.23) | 0.58 | 1.36 (0.68 to 2.73) | 0.38 |
| | No | Reference | | Reference | |

\* $p \leq \cdot 05$ and formatted bold.

\*\* $p \leq \cdot 001$ and formatted bold.

Some results are rounded to 3 rather than 2 decimal places to distinguish between $p \leq \cdot 05$ and $p \leq \cdot 001$.

† When interpretating the predictor listed in the row, we controlled for the other variables listed here: participant gender, age, region, household income, employment status, education level, marital status, ethnicity, and the child's gender and school year.

^^^ Not included variable highly correlated with employment variable.

[1] Working includes students and volunteers.

[2] Question only offered to participants who reported working in a paid job (full-time, part-time, and self-employed) and not to participants who reported being a student, on furlough and a volunteer.

[3] Participants that reported no access to a garden but had access to other outdoor spaces such as patio, terrace, and balcony.

Abbreviations: CI = confidence interval.

There appeared to be a cluster of predictive variables that indicated that children with worse psychological well-being had higher odds of meeting up with non-household family members. Variables in this group included children or their parent having low well-being and children being worried about non-household family members. We interpret these findings as the close contact may have been used to try to improve children's well-being [34], although as we are unable to determine causality; it could be that children's well-being reduced because of meeting their non-household family members.

Children who had extra support before the school closures and had special educational needs also had higher odds of non-household family close contact. This suggests that worries about education also increased the odds of non-household close contact, which aligns with previous research [7].

Our results were less clear in terms of the perceived risk of COVID-19 and the impact on children's non-household family close contact. We did not find any associations between children's non-household family close contact and the lockdown statements that related to COVID-19. This finding is at odds with previous research [7]. In addition, children who were vulnerable to COVID-19 and lived with someone over 70 years old had higher odds of non-household family close contact. These findings are concerning, suggesting increased odds for contagion in these more vulnerable groups.

## Children with a low well-being

Over a quarter (26%) of children reported on in our study had low well-being. A study in Switzerland conducted while schools were closed due to the COVID-19 pandemic found children's well-being and family functioning had reduced compared to before the pandemic, and the psychological impacts were greater for children at risk for neurodevelopmental impairments [35]. Children with special educational needs had four times higher odds of having a lower well-

**Table 5. Binary logistic regression comparing statements about lockdown and associations with children's lower well-being (n = 519).**

| Statement | Level | Unadjusted odds ratio (95%, CI) | P-value | Adjusted odds ratio (95%, CI) † | P-value |
|---|---|---|---|---|---|
| If child goes out, she/he is likely to catch coronavirus | 5-point Likert-scale (1 = strongly agree, 5 = strongly disagree) | 0.63** (0.57 to 0.70) | <0.001 | 0.63** (0.57 to 0.70) | <0.001 |
| If child goes out, she/he is likely to bring coronavirus back into our home | 5-point Likert-scale (1 = strongly agree, 5 = strongly disagree) | 0.63** (0.58 to 0.70) | <0.001 | 0.63** (0.57 to 0.70) | <0.001 |
| Child is keeping up with his/her schoolwork | 5-point Likert-scale (1 = strongly agree, 5 = strongly disagree) | 1.43** (1.31 to 1.57) | <0.001 | 1.46** (1.33 to 1.60) | <0.001 |
| I feel confident helping child with her/his schoolwork | 5-point Likert-scale (1 = strongly agree, 5 = strongly disagree) | 1.34** (1.22 to 1.48) | <0.001 | 1.34** (1.21 to 1.48) | <0.001 |
| I feel supported by child's school | 5-point Likert-scale (1 = strongly agree, 5 = strongly disagree) | 1.26** (1.15 to 1.38) | <0.001 | 1.24** (1.12 to 1.37) | <0.001 |
| I have access to all the resources that child needs to do her/his schoolwork | 5-point Likert-scale (1 = strongly agree, 5 = strongly disagree) | 1.30** (1.18 to 1.43) | <0.001 | 1.31** (1.18 to 1.44) | <0.001 |
| During lockdown, child has learned about important things she/he wouldn't normally learn at school | 5-point Likert-scale (1 = strongly agree, 5 = strongly disagree) | 1.12* (1.01 to 1.24) | 0.02 | 1.13* (1.02 to 1.25) | 0.02 |
| In the past 7 days, child has been bored | 5-point Likert-scale (1 = strongly agree, 5 = strongly disagree) | 0.68** (0.62 to 0.75) | <0.001 | 0.68** (0.61 to 0.75) | <0.001 |
| In the past 7 days, my household has had a regular structure to the day | 5-point Likert-scale (1 = strongly agree, 5 = strongly disagree) | 1.23** (1.12 to 1.36) | <0.001 | 1.23** (1.11 to 1.36) | <0.001 |
| In the past 7 days, child has kept in touch with her/his friends | 5-point Likert-scale (1 = strongly agree, 5 = strongly disagree) | 1.15* (1.05 to 1.25) | 0.01 | 1.67 (0.89 to 3.15) | 0.11 |
| Child is worried about coronavirus | 5-point Likert-scale (1 = strongly agree, 5 = strongly disagree) | 0.60** (0.54 to 0.66) | <0.001 | 0.59** (0.53 to 0.65) | <0.001 |
| In the past 7 days, child has felt upset about not seeing other family members who do not live with us | 5-point Likert-scale (1 = strongly agree, 5 = strongly disagree) | 0.64** (0.59 to 0.70) | <0.001 | 0.63** (0.57 to 0.69) | <0.001 |
| In the past 7 days, I have found it hard to keep up with work or other important commitments | 5-point Likert-scale (1 = strongly agree, 5 = strongly disagree) | 0.56** (0.51 to 0.61) | <0.001 | 0.55** (0.49 to 0.60) | <0.001 |
| In the past 7 days, people in my household have been getting along well | 5-point Likert-scale (1 = strongly agree, 5 = strongly disagree) | 1.47** (1.33 to 1.63) | <0.001 | 1.46** (1.31 to 1.63) | <0.001 |
| I am worried about the financial impact of lockdown measures | 5-point Likert-scale (1 = strongly agree, 5 = strongly disagree) | 0.78** (0.71 to 0.85) | <0.001 | 0.77** (0.70 to 0.85) | <0.001 |
| Before the school closures, child had extra support at school | 5-point Likert-scale (1 = strongly agree, 5 = strongly disagree) | 0.68** (0.63 to 0.74) | <0.001 | 0.69** (0.63 to 0.75) | <0.001 |

\* $p \leq \cdot05$ and formatted bold.

\*\* $p \leq \cdot001$ and formatted bold.

Some results are rounded to 3 rather than 2 decimal places to distinguish between $p \leq \cdot05$ and $p \leq \cdot001$.

† When interpretating the predictor listed in the row, we controlled for the other variables listed here: participant gender, age, region, household income, employment status, education level, marital status, ethnicity, and the child's gender and school year.

Abbreviations: CI = confidence interval.

being compared to children without special educational needs. Previous research supports this finding [36]. In addition, a study suggests that children with special education needs are at risk for emotional, conduct, attention and peer relationship problems [37]. Therefore, the odds of adverse mental health problems could be increased in children with special educational needs due to the increased odds for challenging behaviour during lockdown. Parents who struggle to manage their children's behaviour are prone to using adverse parenting styles, and are at increased risk of family conflict and parental distress [38–40]. That said, 'special educational needs' covers a wide range of health and educational needs and research is needed to unpack what makes these children have higher odds of having a poor well-being; this result could be from educational worries and a lack of academic support for these children [35].

**Table 6. Binary logistic regression comparing reasons for children leaving the home by children with low well-being (n = 519).**

| Number of times that child had left the home for each reason on a continuous scale starting at 0 | Child higher well-being, n, (%) | Child lower well-being, n, (%) | Unadjusted odds ratio (95%, CI) | P-value | Adjusted odds ratio (95%, CI) † | P-value |
|---|---|---|---|---|---|---|
| To go to the shops for groceries, toiletries, or medicines | N = 1491, M = 0.53, SD = 1.03 | N = 519, M = 0.83, SD = 1.24 | **1.25\*\* (1.15 to 1.36)** | <0.001 | **1.24\*\* (1.13 to 1.35)** | <0.001 |
| To go to the shops for other items | N = 1491, M = 0.38, SD = 0.91 | N = 519, M = 0.68, SD = 1.13 | **1.32\*\* (1.20 to 1.45)** | <0.001 | **1.31\*\* (1.19 to 1.45)** | <0.001 |
| To provide help to someone else | N = 1491, M = 0.15, SD = 0.71 | N = 519, M = 0.42, SD = 0.99 | **1.44\*\* (1.27 to 1.64)** | <0.001 | **1.37\*\* (1.21 to 1.56)** | <0.001 |
| To meet friends | N = 1491, M = 0.44, SD = 1.51 | N = 519, M = 0.64, SD = 1.20 | **1.14\*\* (1.06 to 1.24)** | <0.001 | **1.14\* (1.05 to 1.24)** | 0.002 |
| To meet family members who don't live with you | N = 1491, M = 0.42, SD = 0.88 | N = 519, M = 0.59, SD = 0.95 | **1.22\*\* (1.10 to 1.35)** | <0.001 | **1.19\*\* (1.07 to 1.33)** | 0.001 |
| For a medical need (e.g., an outpatient appointment) | N = 1497, M = 0.04, SD = 0.25 | N = 519, M = 0.24, SD = 0.60 | **3.75\*\* (2.75 to 5.11)** | <0.001 | **3.70\*\* (2.64 to 5.18)** | <0.001 |
| To go to school | N = 1491, M = 0.62, SD = 1.48 | N = 519, M = 0.77, SD = 1.51 | 1.07 (1.00 to 1.14) | 0.05 | 1.06 (0.99 to 1.13) | 0.12 |
| For exercise | N = 1491, M = 3.27, SD = 2.99 | N = 519, M = 2.64, SD = 2.53 | **0.92\*\* (0.89 to 0.96)** | <0.001 | **0.93\*\* (0.89 to 0.97)** | <0.001 |
| For another reason | N = 1491, M = 0.15, SD = 0.76 | N = 519, M = 0.33, SD = 0.97 | **1.25\*\* (1.12 to 1.40)** | <0.001 | **1.22\*\* (1.08 to 1.37)** | 0.001 |

\* $p \leq \cdot05$ and formatted bold.

\*\* $p \leq \cdot001$ and formatted bold.

Some results are rounded to 3 rather than 2 decimal places to distinguish between $p \leq \cdot05$ and $p \leq \cdot001$

† When interpretating the predictor listed in the row, we controlled for the other variables listed here: participant gender, age, region, household income, employment status, education level, marital status, ethnicity, and the child's gender and school year.

Abbreviations: N = number; M = mean; SD = standard deviation; % = percentage; CI = confidence interval.

We also observed children had higher odds for low well-being with factors that related to the home environment: parents who reported that their child had lower well-being also tended to report that they were bored or upset about not seeing family members, and that people in the household had not been getting along, or there was no structure to the day. Parents also had higher odds of being worried about the financial impact of lockdown or unable to keep up with work and other commitments. Similar factors were identified in a study conducted during the COVID-19 pandemic that found individuals with poor sleep quality, increased distress due to financial circumstances, dependents, and or who were not adjusting to lockdown had higher odds of experiencing depression [41]. In addition, there is research showing the relationship between parent low well-being and child poor mental health outcomes [5,38,42]. We found that children were at seven times higher odds of lower well-being when the parent had low well-being. Our findings indicate that not only is parental well-being associated with child well-being but there also appears to be a link between parental distress due to home circumstances, such as financial worries.

Factors related to vulnerability to COVID-19 were also associated with children having lower well-being. These findings mirror research showing that factors associated with poor

well-being in children include being worried about a grandparent's vulnerability to COVID-19, infecting their grandparents [15,16] and being vulnerable to COVID-19 themselves [43,44]. Children also had higher odds of a lower well-being if one of their parents was a key worker. Keyworkers commonly interacted with many people daily, which increased their risk of COVID-19 [45]. Families that adopted more protective behaviours because of the risk of COVID-19 also had higher odds of having a child with lower well-being; it is possible that increased levels of protective behaviours reflected a higher general sense of worry about the pandemic within the household.

More positively, children who had a higher well-being had higher odds of having parents who had confidence in home-schooling and who perceived that children were keeping up with their schooling. Research has shown that parental self-efficacy can improve children's well-being [46]. It is possible that this explains our results perhaps by reducing tension within the home about schoolwork. We did not find any associations with parental education, income, or employment status in contrast to previous research [6,13,21,47] although we did note that younger parents had higher odds of reporting low well-being for their children than older parents. Younger adults in general experienced higher levels of stress and anxiety during the pandemic [41] because of more challenging working and living conditions, something which might account for our findings.

Exercise was also a protective factor; children had higher odds of having higher well-being the more times they had left the home to exercise. This finding aligns with previous research that shows the benefits of exercise on physical and mental health [41,48–51]. We were surprised that access to outside space had no associations with either of our outcomes, although having access to outside space does not necessarily mean children use the space. To counterbalance the increased stress on families as a result of the pandemic, exercise should be promoted as a way to maintain well-being and parents should be taught how to increase their self-efficacy in managing difficult situations.

## Limitations

We used a cross-sectional study design and the findings are based on a single point in time. Cross-sectional designs are common in health research as data can be gathered quickly so that the data can be used to respond rapidly to the health threat [52]. We used purposive sampling to meet pre-determined quotas for parent and children characteristics to broadly represent parents and children in England. The use of quota rather than random sampling means that it is not possible to quantify the nature of any bias in the prevalence estimates that we have made. However, we have no reason to believe that the associations between the different variables that we measured would be affected by any theoretical bias [53]. Data were collected from self-reports, which can lead to self-report bias [54]. However, self-report data is a valid study design method [55,56]. The RCADS is designed for children aged between 8 and 18 although parents reported on children from four years old [57]. However, we suggest this has minimal impact on our findings, RCADS has been found to be a reliable and valid measure in children as young as three years old [58].

## Conclusions

During the COVID-19 pandemic we found that although children reduced their close contact, 15% had non-adherent physical contact with non-household family members. The reasons for close contact were largely related to a need for childcare, although factors relating to COVID-19, children's well-being and education were also important. Children who had special educational needs or had a parent with low well-being had higher odds of having a lower well-being

themselves. Exercise and parent self-efficacy with home-schooling may help maintain children's mental and physical health.

## Supporting information

**S1 Text. Full survey material.** The information can be downloaded at: DOI 10.18742/21757232.
(DOCX)

**S1 Appendix.**
(DOCX)

## Author Contributions

**Conceptualization:** Lisa Woodland, Louise E. Smith, Rebecca K. Webster, Richard Amlôt, G. James Rubin.

**Data curation:** Lisa Woodland.

**Funding acquisition:** Rebecca K. Webster, Richard Amlôt.

**Investigation:** Lisa Woodland.

**Methodology:** Lisa Woodland, Louise E. Smith, Samantha K. Brooks, Rebecca K. Webster, Richard Amlôt, Antonia Rubin, G. James Rubin.

**Project administration:** Lisa Woodland.

**Resources:** G. James Rubin.

**Supervision:** Rebecca K. Webster, Richard Amlôt, G. James Rubin.

**Visualization:** Lisa Woodland.

**Writing – original draft:** Lisa Woodland.

**Writing – review & editing:** Louise E. Smith, Samantha K. Brooks, Rebecca K. Webster, Richard Amlôt, Antonia Rubin, G. James Rubin.

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
