## [Decision Letter · Decision Letter 0]

16 Jun 2023

PONE-D-23-01008Children’s interactions with non-household family members and child well-being during the COVID-19 pandemic: a cross-sectional surveyPLOS ONE

Dear Dr. Woodland,

Thank you for submitting your manuscript to PLOS ONE. After careful consideration, we feel that it has merit but does not fully meet PLOS ONE’s publication criteria as it currently stands. Therefore, we invite you to submit a revised version of the manuscript that addresses the points raised during the review process.

We look forward to receiving your revised manuscript.

Kind regards,

Mduduzi Colani Shongwe, PhD

Academic Editor

PLOS ONE

Journal Requirements:

‘I have read the journal's policy and the authors of this manuscript have the following competing interests: G.J.R., R.A. and L.S. participate in the UK’s Scientific Advisory Group for Emergencies, or its subgroups. RA is an employee of the UK Health Security Agency. These groups did not fund the study or authors. The author interests do not alter our adherence to PLOS ONE policies on sharing data and materials.”

We note that one or more of the authors are employed by a commercial company: UK Health Security Agency

Additional Editor Comments:

Dear Dr. Woodland,

I would first like to apologize for the long review process. I had to wait for the 3rd reviewer's comments since Reviewer 1 had recommended acceptance, while Reviewer 2 had recommended a rejection.

While I felt that the article reports important findings, the methodology and results sections have watered them down. Two of the reviewers have raised critical points regarding your article. In fact, Reviewer 2 recommended a rejection. However, I has to carefully consider if this was worth another chance or I should also reject it. After careful consideration, I am of the view of giving you and your team a chance to carefully attend to the points raised by the reviewers and myself and resubmit. The main concern is on the statistical analysis, presentation and interpretation of the results, as you will note in my comments as well as those of the reviewers.

However, note that once you have resubmitted, I will have to subject the revised manuscript to another peer review cycle as I believe the manuscript would have had major changes due to the nature of the recommended suggestions/revisions. If you resubmit the paper without changes in the analysis and interpretation of the results, it will result into a rejection without peer review. 

Academic Editor's Comments:

**Title: **

Since the children were not the respondents in this survey, the title should be rephrased to reflect that. Secondly, substitute “interaction” with “close contact” for consistency. The suggested/rephrased title is: “Perceived children’s close contact with non-household family members and their well-being during the COVID-19 pandemic: a cross-sectional survey”

**Abstract**:

Line 28: Is close contact = interaction? It is not clear. Be consistent throughout the article.Line 32: mention the setting and highest statistical procedure/tests performed.Lines 34-35: Include results of direct reports of interactions/contact by the students as well, not only as perceived by their parents.Line 34: Revisit the statistical language: odds ratios are not a measure of risk, therefore, revise the phrase “more likely” to “had higher odds” throughout the manuscript.Liness 34-40: state the statistical values e.g. aORs and their Cis.Include a sentence that will serve as a “conclusion and recommendation” in the abstract.Include key words after the abstract.

**Introduction**

Do not number the headings for PLOS One (revise the manuscript by strictly following the journal formatting style)Line 54 should be “Box 1”. However, it is not clear what this text appears in a box when it is still the authors presenting their introduction. It would make sense if it was an image from another source document e.g., the media or regulating agency. I suggest this be removed from the box and be presented as part  of the continuous writing of the introduction and be cited as done in the box.Lines 100-104, why these variables, specifically, amongst all other 'risk factors'? What informed this selection?Discuss the known risk factors from previous studies in the introduction section. What factors have other studies identified? Any disagreements/mixed findings on these?Discuss the “well-being” reported in other studies from elsewhere as part of the literature review in the introduction section.

**Methods**:

Lines 106 and 117, BMG stands for?Line 122, if you begin a sentence with a number, write the number in words. Do that throughout the manuscript.Lines 124-125, “… completing the survey quickly…”.   – What was the definition of “quickly”? Who assessed the quickness, BMG or the authors? How did the authors ascertain this?Lines 124-126, indicate to readers whether this practice is common/conventional in such studies i.e. cite previous studies that have used or textbooks that recommend such practice.Line 132, upload the Ethical clearance certificate during the resubmission.Line 134, was the questionnaire/survey administered online or physically? On average, how long did it take participants to complete the survey?Line 147, decide whether to use “interaction” or “close contact”? Be consistent.Line 155, what is the validity of this question in measuring “physical interaction” or close contact?Lines 156-165, discuss the scoring, validity and reliability of the measure. In fact, do that for all the measures in your study. Merge 2.5 with 2.3 (Lines 192-236).Line 179, where were these questions sourced from? Or were they developed for this study? How or informed by?Line 186, Who is the “patient” being referred to here?Lines 192-236, “Kill” this section by merging it with 2.3: analysis should be discussed under 2.6. This will enhance the flow.Lines 197-200, what informed this definition of close contact or interaction?Lines 213-215, recoded from what original categorization? Or from continuous measurement? Specify that for all the variables described as "recoded" below.Lines 229-232, which ones are these now? Are they the same questions referred-to in 2.3.6? If so, merge the sections with 2.3.6. If different, which ones are these now? What questions were asked? How were they developed? What is their validity?Lines 212-236, these should be discussed under 2.3.Line 233, what was the scoring and the possible score range? Merge/discuss this under 2.3.Lines 235-236, what informed this statistical decision? Why were they not valid responses? Is this common statistical practice? If so, site the authorities or sources recommending such a practice. How many of such observations were deemed as missing? Did you assess the impact of removing these observations?Line 236, so, what was the eventual response rate in this study?Line 240, it appears as though hierarchical models would have been the best choice/more efficient here to minimize/control Type I error. Reconsider your analysis decisions/choices and justify your decisions or choices. Otherwise, rerun the analysis using hierarchical models.

**Results:**

Line 248, present the participants' background characteristics first.Line 249-250, you cannot report the findings as though they were stated by the children. In such studies, indicate that the results were reported or perceived by the parents, e.g., here, it must read, ".... 15% of the children were reported to have had ......".Line 256, as indicated in the comments for the abstract section, revisit your statistical language. Using “more likely” is not accurate reporting for odds ratios as they are not a measure of risk. Revise all such words and instead refer to them as "higher odds" or "lower odds" as appropriate. Do that through throughout the results and discussion sections.Ordinarily, this should be a purely descriptive table of the background characteristics of the participants.Throughout the results section, report the ORs and their CIs in text when reporting the results from regression.Indicate whether it is crude or adjusted (i.e Bivariate or multiple). Do that for all tablesShow the "N" for all the tables.Line 269, ordinarily, this should be a purely descriptive table of the background characteristics of the participants.Line 274, it is not accurate to state that all these were adjusted for, yet they were also the predictors. Presenting such a statement implies that there was one or two main independent variable(s) and the rest were control variable/confounders, in which case, you would not interpret them. However, in your study, these were all covariates or predictors, hence, when interpreting one predictor, you should state that you controlled for the rest; not that the model adjusted for all the variables listed. Revist this.Lines, 292-294, the correct presentation should be: "parents aged between 18 and 35 years old, parents who were a key worker and parents with lower well-being had higher odds of reporting that their child had lower well-being." You must do this correction throughout the results section and discussion section.The Tables needs to be simplified to make them easier to read. Present fewer columns, by separately presenting the background characteristics, then descriptives of the variables in Tables 4 and 5 (as Supplementary files/Supporting Information) and collapse Tables 4 and 5 as suggested below.Tables 4 & 5, is it possible to create composite scores of each variable and present them rather than ‘dumping’ all the individual items into the regression models, which is what appears to be the case in the current presentation. That is, these would then be two variables represented each table.

**Discussion**

The discussion section will have to be revised to align with the changes suggested abo

Reviewers' comments:

Reviewer's Responses to Questions

**Comments to the Author**

1. Is the manuscript technically sound, and do the data support the conclusions?

Reviewer #1: Yes

Reviewer #2: Partly

Reviewer #3: Yes

2. Has the statistical analysis been performed appropriately and rigorously? 

Reviewer #1: Yes

Reviewer #2: No

Reviewer #3: Yes

3. Have the authors made all data underlying the findings in their manuscript fully available?

Reviewer #1: Yes

Reviewer #2: Yes

Reviewer #3: Yes

4. Is the manuscript presented in an intelligible fashion and written in standard English?

Reviewer #1: Yes

Reviewer #2: Yes

Reviewer #3: Yes

5. Review Comments to the Author

Reviewer #1: 

The authors have chosen an interesting topic. The concept of the paper is so smart, starting with the introduction, methodology, discussion to a conclusion.In the limitation part of the paper, you better to mention that using the cross sectional survey as the major limitation of this paper.

Reviewer #2: 

Raw data for participant characteristics at baseline? Looking at the survey questions, you have collected a lot of important data but a lot of it is unreported here. I appreciate it may be contained in other publications or future manuscripts, but you need to contextualize your findings.Table 1: check grammar, specify child vs parental age in the column. Add child age: what was the range? This would have had a very significant impact on interpreting results.Did you conduct sub analyses for income as well? The income range and IQR should be reported. Grouping all participants in this way may appear to produce spurious associations. Sub-analyses for each factor (i.e. income, age) would provide more clarity.Other comments about tables in general: the way the data are presented is confusing. For example, you report % across columns but then use reference ranges across rows (86% of male parents completed the survey about children who had not had family interactions, then appears you report an odds ratio calculated relative to female parents?)You state in the limitations section that you could not adjust for bias created by sampling, why was this the case and what happened when these variables were included in the model?What about the incidence of infection? Medical conditions? What was the definition of special needs/ were they all similar?Minor point: participant reimbursement is unclear: what are points?Conclusion that this places children at risk of clinical mental illness is unfounded. Responses to COVID 19 are social phenomena + this suggestion pathologizes a normal response to distressing circumstances arising due to the pandemic.I recommend simplifying the results and even presenting graphs of the rating scales to visualize your findings. The tables are difficult to follow.

Reviewer #3: The authors assessed Children’s interactions with non-household family members and child well-being during the COVID-19 pandemic. The authors used a number of data collection tools to meet their objective. Generally, this is an important study in the area of child protection and well-being. There are few issues that need clarification before the paper is published.

Under results section, I seem to miss data about the participant’s age range (both children and their caregivers).The choice of including parents/ caregivers as participants not the children themselves, should clearly come out.

Methods

There is a disconnection between the data collected, methods used and the main objective of the study. In line with the study’s objectives, it appears to me that; the authors collected more data than what was required. For example, line 186, the authors collected data on: “Patients and public involvement. What did the authors do with data collected under “Statement about the lockdown” Line 179.Line 192: Analysis: recoding variables. This section is very confusing. The authors need to clearly follow the conventional lay out or common phrases used in manuscripts writing. A similar subheading on analysis appears on line 237.Results: Before the authors point their audience to the tables for visualize the results, they should show some significant results in the texts. For example, results under subheading “risk factors” line 389 are missing.

6. PLOS authors have the option to publish the peer review history of their article (what does this mean?). If published, this will include your full peer review and any attached files.

Reviewer #1: **Yes: **Agmas Wassie Abate

Reviewer #2: No

Reviewer #3: No

---

## [Author Response · Author response to Decision Letter 0]

24 Aug 2023

Response: We have updated the manuscript to align with PLOS ONE’s formatting style.

Response: We have provided more information about how the participants provided consent.

‘I have read the journal's policy and the authors of this manuscript have the following competing interests: G.J.R., R.A. and L.S. participate in the UK’s Scientific Advisory Group for Emergencies, or its subgroups. RA is an employee of the UK Health Security Agency. These groups did not fund the study or authors. The author interests do not alter our adherence to PLOS ONE policies on sharing data and materials.”

We note that one or more of the authors are employed by a commercial company: UK Health Security Agency

Response: We have included the funding statement that you have provided and updated the funding statement to include that one of the authors is affiliated with UK Health Security Agency which is an arm of the UK Government. 

Response: We have updated the competing interests statement to include the statement that you have provided.

 Academic Editor's Comments:

Title: 

1. Since the children were not the respondents in this survey, the title should be rephrased to reflect that. Secondly, substitute “interaction” with “close contact” for consistency. The suggested/rephrased title is: “Perceived children’s close contact with non-household family members and their well-being during the COVID-19 pandemic: a cross-sectional survey”

Response: We have changed the title to “Parent-reported child close contact with non-household family members and their well-being during the COVID-19 pandemic: a cross-sectional survey,” to make it clear that it was parents and not children who responded to the survey. 

Abstract:

2. Line 28: Is close contact = interaction? It is not clear. Be consistent throughout the article.

Response: We have updated the manuscript throughout to use “close contact” rather than interaction for clarity. 

3. Line 32: mention the setting and highest statistical procedure/tests performed.

Response: We have added to the abstract that the cross-sectional survey was “completed at any location” by the participants and the statistical tests that we used. 

4. Lines 34-35: Include results of direct reports of interactions/contact by the students as well, not only as perceived by their parents.

Response: Because all data are based on parental report, we do not have these data. We have made it clearer that the findings are from parents' perceptions about their child’s behaviour. 

5. Line 34: Revisit the statistical language: odds ratios are not a measure of risk, therefore, revise the phrase “more likely” to “had higher odds” throughout the manuscript.

Response: We have revised our manuscript throughout to report “had higher odds.”

6. Liness 34-40: state the statistical values e.g. aORs and their Cis.

Response: We have added the adjusted odd ratio and confidence intervals for each result. 

Include a sentence that will serve as a “conclusion and recommendation” in the abstract.

Response: We have added a conclusion and recommendation to the abstract. 

8. Include key words after the abstract.

Response: We have added six key words after the abstract. 

Introduction

9. Do not number the headings for PLOS One (revise the manuscript by strictly following the journal formatting style)

Response: We have updated the manuscript to follow PLOS One’s formatting style.

10. Line 54 should be “Box 1”. However, it is not clear what this text appears in a box when it is still the authors presenting their introduction. It would make sense if it was an image from another source document e.g., the media or regulating agency. I suggest this be removed from the box and be presented as part of the continuous writing of the introduction and be cited as done in the box.

Response: We have removed the box and added the text to follow on the first paragraph in the introduction.

11. Lines 100-104, why these variables, specifically, amongst all other 'risk factors'? What informed this selection?

Response: We have added that the variables we investigated were to chosen because we had conducted a qualitative study that suggested these variables may have an impact on children’s well-being. 

12. Discuss the known risk factors from previous studies in the introduction section. What factors have other studies identified? Any disagreements/mixed findings on these?

Response: We have reported the risk factors that have been identified in previous research.

13. Discuss the “well-being” reported in other studies from elsewhere as part of the literature review in the introduction section.

Response: We have added more detail about well-being from the systematic review that we reported about.

Methods:

14. Lines 106 and 117, BMG stands for?

Response: BMG does not stand for anything – it is the registered name of the company. Information about the company can be found in the reference that we have cited. 

15. Line 122, if you begin a sentence with a number, write the number in words. Do that throughout the manuscript.

Response: We have updated this throughout the manuscript.

16. Lines 124-125, “… completing the survey quickly…”. – What was the definition of “quickly”? Who assessed the quickness, BMG or the authors? How did the authors ascertain this?

Response: We have clarified that BMG screened out the non-eligible participants and that “quickly” was compared against the average and median survey time. 

17. Lines 124-126, indicate to readers whether this practice is common/conventional in such studies i.e. cite previous studies that have used or textbooks that recommend such practice.

Response: We have provided a reference which defines this as a common problem. 

Line 132, upload the Ethical clearance certificate during the resubmission.

Response: We have provided the ethical clearance with the submission. 

Line 134, was the questionnaire/survey administered online or physically? On average, how long did it take participants to complete the survey?

Response: We have reported that the survey was administered online and provided information about how long the survey took to complete. 

20. Line 147, decide whether to use “interaction” or “close contact”? Be consistent.

Response: We have changed all wording to close contact. 

Line 155, what is the validity of this question in measuring “physical interaction” or close contact?

Response: The survey copied the wording of the guidance given by the UK Government to determine close contact which we have reported in the study and provided a reference. But beyond face validity we do not have data on the association between self-reports to this item and observed close contact. We have added to the limitations that the study used self-report data. 

22. Lines 156-165, discuss the scoring, validity and reliability of the measure. In fact, do that for all the measures in your study. 

Response: We have added a validity and reliability section that has references to show the reliability of the RCADS and PHQ-4. We have also reported that our other study questions have face validity, although we do not have psychometric data for them. 

23. Line 179, where were these questions sourced from? Or were they developed for this study? How or informed by?

Response: We have reported that the questions that related to COVID-19 were designed for this study and were based on our previous study, which is the same study that we used to design our predictor variables and have now referenced in the introduction. 

24. Line 186, Who is the “patient” being referred to here?

Response: We have removed “patient” from the sub-heading. 

25. Lines 192-236, “Kill” this section by merging it with 2.3: analysis should be discussed under 2.6. This will enhance the flow.

Response: We have merged the methods section as you have suggested. 

26. Lines 197-200, what informed this definition of close contact or interaction?

Response: We have reported that we used the UK Government’s definition of close contact and referenced this. 

27. Lines 213-215, recoded from what original categorization? Or from continuous measurement? Specify that for all the variables described as "recoded" below.

Response: We had provided the categorization for each question that were originally reported in lines 213-215, this information can also be found in the full survey materials which has been provided as supplementary materials. 

28. Lines 229-232, which ones are these now? Are they the same questions referred-to in 2.3.6? If so, merge the sections with 2.3.6. If different, which ones are these now? What questions were asked? How were they developed? What is their validity?

Response: We suggest that this comment has been resolved from merging the method sections and adding a validity and reliability section in response to previous comments. 

Lines 212-236, these should be discussed under 2.3.

Response: We have merged the method section as you have previously suggested.

30. Line 233, what was the scoring and the possible score range? Merge/discuss this under 2.3.

 Response: We have merged the method section as you have previously suggested.

31. Lines 235-236, what informed this statistical decision? Why were they not valid responses? Is this common statistical practice? If so, site the authorities or sources recommending such a practice. How many of such observations were deemed as missing? Did you assess the impact of removing these observations?

Response: We have added to the manuscript that the reason we adopted this approach for outcome variables was because it was not possible to categorise participants as adherent or not, or as having good wellbeing or not. For predictors we adopted this approach to provide a clearer understanding of the difference between endorsing, or not endorsing, each variable in terms of its impact on our outcomes.

32. Line 236, so, what was the eventual response rate in this study?

Response: Quota sampling is a non-probability based approach. Response rates are therefore not helpful – the rates for each quota will differ, the denominator for each is essentially is unknown, and the predominant source of bias is in the make up of the underlying panels that are being recruited from, rather than whether the sample is representative of the panel. We have now reminded readers about this in the paper and given a reference for further reading. 

33. Line 240, it appears as though hierarchical models would have been the best choice/more efficient here to minimize/control Type I error. Reconsider your analysis decisions/choices and justify your decisions or choices. Otherwise, rerun the analysis using hierarchical models.

Response: We agree with the reviewer that using hierarchical models would have helped minimise Type I error. The aim of this study was to investigate independent factors that may have been associated with outcomes, so that we could make specific suggestions to public health agencies for potential targets of communications or policy interventions. We based our inclusion of predictive factors on previous research (published and conducted by our group). Factors investigated were wide ranging, from parent and child personal and clinical characteristics, to parent perceptions of COVID-19, home-schooling, child perceptions of COVID-19, coping in the last week, and number of times having left the home. If we had used hierarchical models, we would have been investigating the effect of a factor, while controlling for all other factors. We are not sure whether e.g., parental perceptions about home schooling are associated with the number of outings in the last week. Therefore, we did not feel comfortable including all factors in a single statistical model. Instead, we chose to control only for parent and child personal characteristics, as these are things that are not modifiable. Furthermore, there is no theoretical basis for the order of inclusion of factors in blocks, if we were to use hierarchical models. For these reasons, we chose to conduct analyses as described in the manuscript and did not use hierarchical models.

Results:

34. Line 248, present the participants' background characteristics first.

Response: We have reported some parent characteristics narratively in the opening paragraph and added a table reporting parent and child characteristics. 

35. Line 249-250, you cannot report the findings as though they were stated by the children. In such studies, indicate that the results were reported or perceived by the parents, e.g., here, it must read, ".... 15% of the children were reported to have had ......".

Response: We have added “perceived by the parents” throughout the results to make it clear that it was not the children who responded to the study. 

Line 256, as indicated in the comments for the abstract section, revisit your statistical language. Using “more likely” is not accurate reporting for odds ratios as they are not a measure of risk. Revise all such words and instead refer to them as "higher odds" or "lower odds" as appropriate. Do that through throughout the results and discussion sections.

Response: We have amended as you have suggested. 

Ordinarily, this should be a purely descriptive table of the background characteristics of the participants.

Response: The tables provided all of the details necessary to understand the association between the predictor and outcome variables. However, we have now changed this so that we have one table that only reports parent and child characteristics and associations are reported in two another tables. 

38. Throughout the results section, report the ORs and their CIs in text when reporting the results from regression.

Response: These are listed in the tables, and we felt repeating the information in the text would be duplicative, but we have now made this change as requested. 

39. Indicate whether it is crude or adjusted (i.e Bivariate or multiple). Do that for all tables

Response: The tables already show both adjusted and unadjusted associations, although we have added “unadjusted” odds ratio to make this clearer. 

Show the "N" for all the tables.

Response: The tables already show the sample sizes and percentages. For the continuous variables we have also reported the mean and standard deviation. We have added abbreviations to the footnotes of each table to make this clearer. 

41. Line 269, ordinarily, this should be a purely descriptive table of the background characteristics of the participants.

Response: We have amended how the tables are presented following reviewer comments. 

42. Line 274, it is not accurate to state that all these were adjusted for, yet they were also the predictors. Presenting such a statement implies that there was one or two main independent variable(s) and the rest were control variable/confounders, in which case, you would not interpret them. However, in your study, these were all covariates or predictors, hence, when interpreting one predictor, you should state that you controlled for the rest; not that the model adjusted for all the variables listed. Revisit this.

Response: We have changed the footnote to state that “when interpreting the predictor listed in the row, we controlled for the other variables listed” as you have suggested. 

43. Lines, 292-294, the correct presentation should be: "parents aged between 18 and 35 years old, parents who were a key worker and parents with lower well-being had higher odds of reporting that their child had lower well-being." You must do this correction throughout the results section and discussion section.

Response: We have updated the results section as you have suggested. 

44. The Tables needs to be simplified to make them easier to read. Present fewer columns, by separately presenting the background characteristics, then descriptives of the variables in Tables 4 and 5 (as Supplementary files/Supporting Information) and collapse Tables 4 and 5 as suggested below.

Response: We have separated the tables and the outcome frequencies are reported in the appendix to reduce the number of columns that are presented in the tables in the main text. 

45. Tables 4 & 5, is it possible to create composite scores of each variable and present them rather than ‘dumping’ all the individual items into the regression models, which is what appears to be the case in the current presentation. That is, these would then be two variables represented each table.

Response: For similar reasons to why we did not use hierarchical models, the study aims to investigate the independent associations between the predictor variables that we reported about and our two outcomes. Therefore, a factor analysis or similar method to create a composite score of similar variables did not suit our study’s aims, by combining variables we would not be able to investigate the independent effects of each predictor variable on parents’ perception of their children’s close contact behaviour and well-being. By doing so now would also mean that we would be revising our analytic strategy after reviewing our data. In addition, future studies will more easily be able to replicate single items rather than a composite score as they can choose the variables that they want to investigate. 

Discussion

The discussion section will have to be revised to align with the changes suggested above.

Response: We have provided the rationale for why we have not altered our analysis strategy as suggested above. Therefore, we suggest that the discussion appropriately discusses our findings.

Reviewer #1: 

1. The authors have chosen an interesting topic. The concept of the paper is so smart, starting with the introduction, methodology, discussion to a conclusion.

2. In the limitation part of the paper, you better to mention that using the cross sectional survey as the major limitation of this paper.

Response: We have added that we used a cross-sectional design to our limitations. 

 Reviewer #2: 

1. Raw data for participant characteristics at baseline? Looking at the survey questions, you have collected a lot of important data but a lot of it is unreported here. I appreciate it may be contained in other publications or future manuscripts, but you need to contextualize your findings.

Response: We are not clear what the reviewer means by “baseline” – this was a cross-sectional survey or the data that they indicate we have not reported. However, we do now report a separate table of participant characteristics as suggested by the editor. 

2. Table 1: check grammar, specify child vs parental age in the column. Add child age: what was the range? This would have had a very significant impact on interpreting results.

Response: We have added ‘parent’ or ‘child’ to the gender, age and school year variables description in the table to make this clearer. The eligibility criteria states the age of the children who were included in the study (“were a parent or guardian to a school-aged child (4-18 years old”)). The table that reports parent and child characteristics that we have created following reviewer comments reports the parents age group and the child’s Key Stage. We have also added the child’s age range to this table to make the child’s age clearer. 

3. Did you conduct sub analyses for income as well? The income range and IQR should be reported. Grouping all participants in this way may appear to produce spurious associations. Sub-analyses for each factor (i.e. income, age) would provide more clarity.

Response: We agree that you can code variables in various different ways and each method may produce different results. Therefore, the decisions that we made about how we would code our variables were made carefully based on previous research and our data before we conducted our analysis. The number of levels in a variable was a main driver, we tried to balance not having too many levels as not to increase the risk of type 2 errors with being able to identify the groups most at risk. There were 15 different income brackets and one “prefer not to say,” we decided to create a binary variable based on the mean household income in our data (£40,000 to £44,000 bracket) and the mean UK household income in the UK (£32,000). Providing the IQR for income would not be useful as this was a categorical variable. We have conducted sensitivity analysis about this variable, which show a similar pattern to the findings that we have presented. 

We decided to recode age, into three levels based on the Mean = 43 years of age; 25th percentile = 37 years of age; and 75th percentile = 49 years of age and with trying to group the income brackets evenly (by number of responses and income range). Previous research also indicated that very young or old parents may have difference impacts on our outcomes compared to middle aged parents and therefore, keeping the age variable as a continuous variable did not align to our study’s aims. We did not run sensitivity analysis on these data as increasing the number of levels in the age variable, by increments of 10 years for example produces low cell counts in several categories and therefore we are unable to analyse the data. 

4. Other comments about tables in general: the way the data are presented is confusing. For example, you report % across columns but then use reference ranges across rows (86% of male parents completed the survey about children who had not had family interactions, then appears you report an odds ratio calculated relative to female parents?)

Response: The reviewer has correctly interpreted how we had presented the table. However, we have now split them up into multiple tables at the suggestion of the editor so that the tables in the main manuscript have fewer columns for clarity.

5. You state in the limitations section that you could not adjust for bias created by sampling, why was this the case and what happened when these variables were included in the model?

Response: We are not certain we understand the reviewer’s point here. We used quota, rather than random sampling. This might lead to bias. There is no analysis we can do to adjust for this – the bias is inherent in the sampling strategy we used. The best we can do is to be honest with readers about the limitation.

6. What about the incidence of infection? Medical conditions? What was the definition of special needs/ were they all similar?

Response: We did not assess the incidence of infection, rather parents’ perceptions about their or their child’s perceptions of COVID-19 (e.g., child is worried about COVID-19 and if child goes out, she/he is likely to catch COVID-19) to identify how these perceptions may or may not influence children’s contact behaviour and well-being. Similarly, the importance of the study was not the type of medical conditions that the child or household member had that made them vulnerable to COVID-19 and if the child had special educational needs, rather whether or not these perceptions influenced behaviour. We asked the parent “does the [child] have special educational needs,” which we also discuss in the discussion: “special educational needs covers a wide range of health and educational needs and research is needed to unpack what makes these children most at risk of poor well-being.” 

7. Minor point: participant reimbursement is unclear: what are points?

Response: Panel members earn points for each survey they complete. They can then exchange their accumulated points for rewards once they have accumulated enough (that is, once they have completed enough surveys). For our survey, they earned points worth about 60p. We have added this to the manuscript. 

8. Conclusion that this places children at risk of clinical mental illness is unfounded. Responses to COVID 19 are social phenomena + this suggestion pathologizes a normal response to distressing circumstances arising due to the pandemic.

Response: We have removed this statement from the discussion. 

9. I recommend simplifying the results and even presenting graphs of the rating scales to visualize your findings. The tables are difficult to follow.

Response: We have revised the tables in response to previous comments, which have simplified the tables and we suggest that graphs may not add to the findings. 

 Reviewer #3: The authors assessed Children’s interactions with non-household family members and child well-being during the COVID-19 pandemic. The authors used a number of data collection tools to meet their objective. Generally, this is an important study in the area of child protection and well-being. There are few issues that need clarification before the paper is published.

1. Under results section, I seem to miss data about the participant’s age range (both children and their caregivers).

Response: We have added a Table that presents the parent and child’s characteristics, which includes parents age range and children’s Key Stage. We have also tried to make it clear in the tables whether the variables are referring to the parents or their children in response to your previous comment. 

2. The choice of including parents/ caregivers as participants not the children themselves, should clearly come out.

Response: We have altered the wording throughout to make it clear that the study participants are parents and they are reporting about their perceptions of their children’s behaviour. 

 Methods

1. There is a disconnection between the data collected, methods used and the main objective of the study. In line with the study’s objectives, it appears to me that; the authors collected more data than what was required. For example, line 186, the authors collected data on: “Patients and public involvement. What did the authors do with data collected under “Statement about the lockdown” Line 179.

Response: We have changed the structure of the method and materials, which has made this section clearer, including that we did not collect any data about patients and the public, rather they were used to pilot our survey questions; and that the statements about the lockdown are reported in Tables 3 and 5. 

2. Line 192: Analysis: recoding variables. This section is very confusing. The authors need to clearly follow the conventional lay out or common phrases used in manuscripts writing. A similar subheading on analysis appears on line 237.

Response: As stated in the previous comment, we have revised the layout of our materials and methods section.

3. Results: Before the authors point their audience to the tables for visualize the results, they should show some significant results in the texts. For example, results under subheading “risk factors” line 389 are missing.

Response: We have narratively reported the significant results before we present the full tabulated results. We have also added the adjusted odd ratios to the narrative results. Line 389 in the original manuscript refers to the discussion section that states: “attention and peer relationship problems [25]. Therefore, the risk of adverse mental...,” we are unsure what section is missing. If this comment has not been resolved in our revised manuscript, we welcome further clarification on this comment.

---

## [Decision Letter · Decision Letter 1]

10 Sep 2023

PONE-D-23-01008R1Parent-reported child’s close contact with non-household family members and their well-being during the COVID-19 pandemic: a cross-sectional surveyPLOS ONE

Dear Dr. Woodland,

Thank you for submitting your manuscript to PLOS ONE. After careful consideration, we feel that it has merit but does not fully meet PLOS ONE’s publication criteria as it currently stands. Therefore, we invite you to submit a revised version of the manuscript that addresses the points raised during the review process.

We look forward to receiving your revised manuscript.

Kind regards,

Mduduzi Colani Shongwe

Academic Editor

PLOS ONE

Additional Editor Comments (if provided):

Thank you for providing the clarification/justification for the analytic methods you chose for your paper. It is now clearer to me how you approached the analysis: you were examining the effects of one predictor while adjusting for the characteristics you mentioned i.e. you had one independent variable with its confounders being controlled for at a time, hence, you should not interpret the effects of those confounders to avoid Table 2 fallacy. 

However, if that was the approach, it means there must be an explanation or demonstration of how the confounders (control variables) were selected for each multivariable model. I suggest you do the following:

Explain in the "Data analysis" section of the methods how confounders (control variables) for each model were selected for the different outcomes (separately for outcome 1 and outcome 2). Submit a supplementary file/Appendix Tables demonstrating how confounders were selected for each outcome and for each multivariable model since, theoretically, the association of each independent variable (with each outcome) will have its own confounders, which may not be the same across. If they were the same across models and outcomes (which is not expected), then that must also be demonstrated in the supplementary file. If you can demonstrate this part, I believe I will be in a position to accept the paper without subjecting it to a review cycle.In Table 2 and Table 4, do not show the results of the confounders as you should not interpret them, instead maintain the statement in the table footnotes which says: "† When interpreting the predictor listed in the row, we controlled for the other variables listed here: participant gender, age, region, household income, employment status, education level, marital status, ethnicity, and the child’s gender and school year"Delete the sentences in the result section in Lines 350-352; 40-403; and 560-566 , because you would have not shown those results, assuming you will implement the recommendation in point 3 above. This will also ensure that you do not interpret the confounders to avoid the Table 2 fallacy I mentioned above.Discussion: you did not measure risk in this study; you measured the odds, correct that throughout the discussion section as previously suggested.Line 600, the keywords are misplaced, they should appear under the abstract, not here.Replace Table 7 and Table 10 with tables showing how confounders were selected.

Reviewers' comments:

Reviewer's Responses to Questions

**Comments to the Author**

1. If the authors have adequately addressed your comments raised in a previous round of review and you feel that this manuscript is now acceptable for publication, you may indicate that here to bypass the “Comments to the Author” section, enter your conflict of interest statement in the “Confidential to Editor” section, and submit your "Accept" recommendation.

Reviewer #3: All comments have been addressed

2. Is the manuscript technically sound, and do the data support the conclusions?

Reviewer #3: Yes

3. Has the statistical analysis been performed appropriately and rigorously? 

Reviewer #3: Yes

4. Have the authors made all data underlying the findings in their manuscript fully available?

Reviewer #3: Yes

5. Is the manuscript presented in an intelligible fashion and written in standard English?

Reviewer #3: Yes

6. Review Comments to the Author

Reviewer #3: Thank you for according me this wonderful opportunity to review this manuscript. In general, the authors have adequately responded to my comments. I recommend that the Manuscript be publication in your journal.

7. PLOS authors have the option to publish the peer review history of their article (what does this mean?). If published, this will include your full peer review and any attached files.

Reviewer #3: No

---

## [Author Response · Author response to Decision Letter 1]

18 Sep 2023

Additional Editor Comments (if provided):

Thank you for providing the clarification/justification for the analytic methods you chose for your paper. It is now clearer to me how you approached the analysis: you were examining the effects of one predictor while adjusting for the characteristics you mentioned i.e. you had one independent variable with its confounders being controlled for at a time, hence, you should not interpret the effects of those confounders to avoid Table 2 fallacy. 

However, if that was the approach, it means there must be an explanation or demonstration of how the confounders (control variables) were selected for each multivariable model. I suggest you do the following:

1. Explain in the "Data analysis" section of the methods how confounders (control variables) for each model were selected for the different outcomes (separately for outcome 1 and outcome 2). 

Response: We recognise there are different approaches to the selection of confounders. Our approach, which we have now made explicit in our manuscript, was to control for all ‘pre-exposure’ variables (i.e. demographics and related variables). A citation has now been provided to a key paper on selection of confounders to confirm that this is a reasonable approach and also to allow the reader to judge this for themselves. Our data will also be publicly available (upon publication) and readers are, of course, at liberty to use our data to test alternative approaches to confounder selection. 

VanderWeele, T. J., Principles of confounder selection. European journal of epidemiology 2019, 34, 211-219.

2. Submit a supplementary file/Appendix Tables demonstrating how confounders were selected for each outcome and for each multivariable model since, theoretically, the association of each independent variable (with each outcome) will have its own confounders, which may not be the same across. If they were the same across models and outcomes (which is not expected), then that must also be demonstrated in the supplementary file. If you can demonstrate this part, I believe I will be in a position to accept the paper without subjecting it to a review cycle.

Response: As per our previous response, we acknowledge that data can be analysed in a variety of ways. We have made it clear throughout the manuscript that our confounders were chosen based on a theoretical basis; research has suggested that parent and child characteristics may have an influence on both of our outcomes, although we wanted to investigate how these associations, independently and as covariates impacted the outcome. These analysis decisions are explained throughout our manuscript following the changes that we have made during this review process, for which we are grateful. 

3. In Table 2 and Table 4, do not show the results of the confounders as you should not interpret them, instead maintain the statement in the table footnotes which says: "† When interpreting the predictor listed in the row, we controlled for the other variables listed here: participant gender, age, region, household income, employment status, education level, marital status, ethnicity, and the child’s gender and school year"

Response: We appreciate that at times it may not be appropriate to report the confounders, although as we have explained previously, these results are important in order to explore our research question. We made several changes to the table in the previous peer review round, and we welcome further comments about presentation, although we suggest that the results of the associations of demographics with the outcomes of interest are relevant and that a proper consideration of this (and one that is easy to explain and understand) requires us to adjust for other demographic variables. This was our pre-planned analytic strategy. 

4. Delete the sentences in the result section in Lines 350-352; 40-403; and 560-566, because you would have not shown those results, assuming you will implement the recommendation in point 3 above. This will also ensure that you do not interpret the confounders to avoid the Table 2 fallacy I mentioned above.

Response: Thank you for this comment, although we have described in the previous comment how important these results are to the literature and warrant reporting on. 

5. Discussion: you did not measure risk in this study; you measured the odds, correct that throughout the discussion section as previously suggested.

Response: Apologies, we have updated this.

6. Line 600, the keywords are misplaced, they should appear under the abstract, not here.

Response: We have removed the keywords from this section. 

7. Replace Table 7 and Table 10 with tables showing how confounders were selected.

Response: We suggest that this comment has been addressed in our earlier responses. 

If the authors have adequately addressed your comments raised in a previous round of review and you feel that this manuscript is now acceptable for publication, you may indicate that here to bypass the “Comments to the Author” section, enter your conflict of interest statement in the “Confidential to Editor” section, and submit your "Accept" recommendation.

Reviewer #3: All comments have been addressed

Reviewer #3: Thank you for according me this wonderful opportunity to review this manuscript. In general, the authors have adequately responded to my comments. I recommend that the Manuscript be publication in your journal.

Response: We are pleased that we have addressed all of the reviewers’ comments and that they both believe it is ready for publication. We thank them for their time and the comments that they have provided.

---

## [Editor Report · Decision Letter 2]

20 Sep 2023

Parent-reported child’s close contact with non-household family members and their well-being during the COVID-19 pandemic: a cross-sectional survey

PONE-D-23-01008R2

Dear Dr. Woodland,

We’re pleased to inform you that your manuscript has been judged scientifically suitable for publication and will be formally accepted for publication once it meets all outstanding technical requirements.

Kind regards,

Mduduzi Colani Shongwe, PhD

Academic Editor

PLOS ONE

Additional Editor Comments (optional):

Thank you for responding to the Round 2 of comments. While I may not agree 100% with some of the methodological choices in your paper, I agree that the review has improved the manuscript tremendously and has afforded your team a chance to explain all methodological and statistical decisions so that readers understand why you made the choices you made.

However, may I clarify this statement: "We are pleased that we have addressed all of the reviewers’ comments and that they both believe it is ready for publication. We thank them for their time and

the comments that they have provided." - One of the reviewers actually recommended for a rejection during the first review cycle and in the second round, she declined to review this manuscript without giving any reasons. So, for the records, it is not accurate that they "... both believe it is ready for publication."
---

## [Editor Report · Acceptance letter]

11 Oct 2023

PONE-D-23-01008R2 

Parent-reported child’s close contact with non-household family members and their well-being during the COVID-19 pandemic: a cross-sectional survey 

Dear Dr. Woodland:

I'm pleased to inform you that your manuscript has been deemed suitable for publication in PLOS ONE. Congratulations! Your manuscript is now with our production department. 

Kind regards, 

on behalf of

Dr. Mduduzi Colani Shongwe 

Academic Editor

PLOS ONE